# Selina-1,3,7(11)-trien-8-one and Oxidoselina-1,3,7(11)-trien-8-one from *Eugenia uniflora* Leaf Essential Oil and Their Cytotoxic Effects on Human Cell Lines

**DOI:** 10.3390/molecules26030740

**Published:** 2021-01-31

**Authors:** Jociani Ascari, Marcos Felipe Maciel Pereira, Vinicius Monteiro Schaffka, Domingos Sávio Nunes, Cássia Gonçalves Magalhães, Jânio Sousa Santos, Daniel Granato, Mariana Araújo Vieira do Carmo, Luciana Azevedo, Marcos Vinicio Lopes Rodrigues Archilha, Dilamara Riva Scharf

**Affiliations:** 1Department of Biology, Federal Technological University of Paraná, 85892-000 Santa Helena, PR, Brazil; marcosfmpereira@gmail.com; 2Department of Chemistry, State University of Ponta Grossa, 84030-900 Ponta Grossa, PR, Brazil; vschaffka@gmail.com (V.M.S.); senunsd@gmail.com (D.S.N.); cassiagmag@yahoo.com.br (C.G.M.); santosjs.food@gmail.com (J.S.S.); 3Food Processing and Quality, Natural Resources Institute Finland (Luke), FI-02150 Espoo, Finland; 4Nutrition Faculty, Federal University of Alfenas, 37130-000 Alfenas, MG, Brazil; marianavieira06@hotmail.com (M.A.V.d.C.); lucianaazevedo2010@gmail.com (L.A.); 5Institute of Chemistry, University of São Paulo, 05508-000 São Paulo, SP, Brazil; archilha@iq.usp.br; 6Department of Chemical Engineering, Regional University of Blumenau Foundation, 89030-000 Blumenau, SC, Brazil; driva@furb.br

**Keywords:** *Eugenia uniflora*, selina-1,3,7(11)-trien-8-one, oxidoselina-1,3,7(11)-trien-8-one, cytotoxicity

## Abstract

The sesquiterpenes selina-1,3,7(11)-trien-8-one and oxidoselina-1,3,7(11)-trien-8-one were isolated from the essential oil of *Eugenia uniflora* L. leaves. The structures were elucidated using spectrometric methods (UV, GC–MS, NMR, and specific optical rotation). The relationship between antioxidant activity, as determined by DPPH assay, and the cytotoxic effect was evaluated using tumor cells, namely lung adenocarcinoma epithelial cells (A549) and human hepatoma carcinoma cells (HepG2), as well as a model of normal human lung fibroblast cells (IMR90). Both compounds did not show prominent free-radical scavenging activity according to DPPH assay, and did not inhibit lipid peroxidation in Wistar rat brain homogenate. The isolated compounds showed pro-oxidative effects and cytotoxicity in relation to the IMR90 cell line.

## 1. Introduction

*Eugenia uniflora* L. (Myrtaceae), popularly known in Brazil as pitangueira, is a small tree species widely distributed in South and Central America [1,2]. In view of the pharmacological activities attributed to the species in folk medicine, the plant has received intense scientific attention and become of interest to the cosmetic industry [3,4,5]. *E. uniflora* currently has at least 41 recognized and registered heterotypic synonyms in Brazil [6], which indicates the existence of a large number of varieties for this plant species.

In the first isolation reports of volatile compounds from *E. uniflora* leaves, some volatile sesquiterpenes were obtained, such as furanodiene, isofuranoelemene, curzerene, germacrene B, γ-elemene, selina-4(14),7(11)-diene, β-elemene, caryophyllene, germacrene D, bicyclogermacrene, β-selinene, atractylone, germacrone, selina-1,3,7(11)-trien-8-one (**1**), and oxidoselina-1,3,7(11)-trien-8-one (**2**) [7,8]. Most of these mentioned sesquiterpenes are also known as volatile components of various plant species from other botanical families, especially from the Zingiberaceae (*Curcuma*), Burseraceae (*Commiphora*), and Apiaceae (*Smyrnium*) [9,10,11,12,13]. To date, the sesquiterpenes selina-1,3,7(11)-trien-8-one (**1**) and oxidoselina-1,3,7(11)-trien-8-one (**2**) (Figure 1) have been found solely in *E. uniflora*.

Studies on the chemical composition of essential oils from *E. uniflora* leaves show that compounds **1** and **2** are not thermosensitive, but some other components can undergo Cope rearrangement transformations during gas chromatography analysis, a method that is proven to be especially inadequate for the analysis of sesquiterpenes such as furanodiene, curzerene, germacrene B, γ-elemene, germacrone, and β-elemenone [7,8]. This analytical aspect requires attention when interpreting the results of numerous publications on the chemical composition and biological activities of essential oils.

The chemical composition of the volatiles obtained from different *E. uniflora* chemotypes shows they are sesquiterpenoids, including compounds **1** and **2**, germacrenes A, B, and D, germacrone, caryophyllene, atractylone, curzerene, and furanodiene, with some of these having been cited as chemical markers for the species [9,10,11,12].

Several studies have shown the potential biological activities of *E. uniflora* essential oils. Costa (2010) demonstrated the antifungal activity of essential oils from *E. uniflora* leaves against *Paracoccidioides brasiliensis* for samples presenting curzerene, germacrene D, and germacrene A as the major GC constituents [10]. An essential oil from the leaves of *E. uniflora*, containing **1** and **2** as the major compounds, showed antifungal activity against standard strains of *Candida albicans*, *C. krusei*, and *C. tropicalis* [13]. Additionally, another essential oil from *E. uniflora* leaves presenting curzerene and high proportions of **1** and **2** according to GC analyses showed relevant hepatoprotective activity, and the therapeutic potential of both compounds for the development of herbal medicines was mentioned [14,15].

Studies evaluating the cytotoxic action of *E. uniflora* essential oils on cancer cell lines have shown promising results. Ogunwande (2005) demonstrated that an essential oil presenting the sesquiterpene **1**, curzerene, atractylone, and furanodiene has cytotoxic and antiproliferative effects in relation to prostatic carcinoma cells (PC-3), hepatocellular carcinoma cells (HepG2), and breast ductal carcinoma cells (578T) [16]. Figueiredo (2019) evaluated the cytotoxic effects of *E. uniflora* oils and curzerene in relation to human colon (HCT-116), gastric (AGP-01), melanoma (SKMEL-19), and normal human fibroblast (MRC-5) cell lines. Curzerene showed more significant activity against SKMEL-19 cells, with oil samples that contained **1** and **2** in addition to curzerene, germacrene B, caryophyllene oxide, caryophyllene, β-elemene, and γ-elemene as the major compounds showing cytotoxic activity against all cell lines [12].

Given the importance of volatile compounds as bioactive natural products from medicinal and food plants along with the need to track their different biological activities, two sesquiterpenes from the leaves of *E. uniflora*, selina-1,3,7-trien-8-one (**1**) and oxidoselina-1,3,7(11)-trien-8-one (**2**), were isolated using conventional column chromatography. Structural elucidation was performed using modern spectroscopy and specific optical rotation techniques. We also present, here, the results of antioxidant analyses for the two sesquiterpenes and an assessment of the relationship between antioxidant power and cytotoxic effects in different normal and cancerous human cells.

## 2. Results and Discussion

The chemical composition found in this work for the *E. uniflora* leaf oil is shown in Table 1. The sample of the essential oil studied here presents the striking profile of one of the best known chemotypes of this plant species [14,15,16,17,18], in which the four main components stand out, namely curzerene, germacrene B, and **1** and **2**. In fact, the GC–MS analytical chromatogram of the essential oil obtained in the present work is even very similar to that obtained from leaves of the tree cultivated in Nigeria, which were used in a chemical study by Weyerstahl (1988) [8].

Chemical transformations of several of the components of the *E. uniflora* leaf essential oil are known to be caused by the heating conditions commonly used in GC–MS analysis, and a cold method analysis is required for correct composition evaluation [7,8]. Particularly important for the case of the sample described in the present research are the transformations of germacrene B into γ-elemene, furanediene into curzerene, and germacrone into β-elemenone, noting that compounds **1** and **2** are stable under the heating conditions used in GC–MS.

Weyerstahl and collaborators (1988) first isolated and identified compounds **1** and **2** as the major components of the essential oil, and during the decades that followed, these sesquiterpenes were found to also occur in various *E. uniflora* chemotypes [8,9,10,11,12]. Compound **1** was isolated in our laboratory as an oil and presented optical activity ${\left[\alpha \right]}_{D}^{20}-8{}^{\circ}$ (*c* 1.0, CHCl_3_), very close to the value registered in the literature when the compound was isolated for the first time [8]. Compound **1**, obtained by synthesis, presented the value of ${\left[\alpha \right]}_{D}^{20}-258{}^{\circ}$ (*c* 1.0, CHCl_3_) [19], which can be now considered as a very discrepant value when compared to that found for the natural substance.

The ^1^H and ^13^C NMR signals of **1** (see Table 2) were assigned based on thorough analysis of COSY, HSQC, and HMBC spectra presented in the Appendix A. The ^1^H NMR spectrum showed signals at *δ*_H_ 2.47 (d, *J* = 14.5 Hz) and *δ*_H_ 2.30 (d, *J* = 14.5 Hz) referring to diastereotopic hydrogens H-9 and H-9’. At *δ*_H_ 5.76 (dd, *J* = 9.4, 5.3 Hz), a double doublet signal corresponds to the methynic hydrogen H-2. Four methyl groups were observed with displacements at *δ*_H_ 1.79 (s, H-12 or H-13), *δ*_H_ 1.94 (d, *J* = 1.6 Hz, H-12 or H-13), *δ*_H_ 1.04 (s, H-14), and at *δ*_H_ 1.85 (s, H-15).

The ^13^C spectrum showed a signal at *δ*_C_ 203.9 for carbonyl carbon and four methyl carbons at *δ*_C_ 21.7 (C-12 or C-13), *δ*_C_ 22.6 (C-12 or C-13), *δ*_C_ 26.7 (C-14), and *δ*_C_ 22.2 (C-15) as well as four methynic carbons at *δ*_C_ 131.7 (C-1), *δ*_C_ 123 (C-2), *δ*_C_ 118 (C-3), and *δ*_C_ 46.1 (C-5). The chemical shifts *δ*_C_ 123.0 (C-2), *δ*_C_ 118.0 (C-3), *δ*_C_ 22.6 (C-12 or C-13), and *δ*_C_ 22.2 (C-15) are significantly different when compared to those reported in the literature for the synthetic compound **1** [19], *δ*_C_ 118 (C-2), *δ*_C_ 123 (C-3), *δ*_C_ 22.2 (C-12 or C-13), and *δ*_C_ 22.6 (C-15) [19]. With COSY, it was possible to correlate the couplings between the signals at *δ*_H_ 5.33 (d, *J* = 9.4 Hz, H-3) with *δ*_H_ 5.76 (d, *J* = 9.4, 5.3 Hz, H-2) and *δ*_H_ 2.66 (dd, *J* = 10.6, 4.8 Hz, H-6) with *δ*_H_ 2.24 (m, H-6’) and *δ*_H_ 2.00 (dd, *J* = 10.6, 4.8 Hz, H-5). The HMBC spectrum analysis showed a correlation of C-7 (*δ*_C_ 132.6) with H-6 (*δ*_H_ 2.66), H-6’ (*δ*_H_ 2.24), H-12 or H-13 (*δ*_H_ 1.79), and H-12 or H-13 (*δ*_H_ 1.94). The correlation of C-3 (*δ*_C_ 118.0) with H-15 (*δ*_H_ 1.85) and C-9 (*δ*_C_ 53.4) with H-14 (*δ*_H_ 1.04) confirmed the positions of the H-14 and H-15 methyl groups. The position of the carbonyl group C-8 (*δ*_C_ 203.9) correlated with the diastereotopic hydrogens H-6 (*δ*_H_ 2.66) and H-6’ (*δ*_H_ 2.24) as well as H-9 (*δ*_H_ 2.47) and H-9’ (*δ*_H_ 2.30).

Compound **2**, also isolated as an oil, presented optical activity ${\left[\alpha \right]}_{D}^{20}-144{}^{\circ}$ (*c* 1.0, CHCl_3_) slightly above that found earlier in the first isolation, ${\left[\alpha \right]}_{D}^{20}-107{}^{\circ}$ (*c* 0.6, CHCl_3_) [8].

The ^1^H and ^13^C NMR signals are shown in Table 2. The ^1^H NMR spectrum showed signals at *δ*_H_ 2.40 (d, *J* = 15.0 Hz) and *δ*_H_ 2.19 (d, *J* = 13.7 Hz) for the diastereotopic H-9 and H-9’ hydrogens. At *δ*_H_ 6.02 (d, *J* = 7.7 Hz) and at *δ*_H_ 6.13 (s) are the signals corresponding to the methynic hydrogens H-2 and H-3. Four methyl groups were observed at the following displacements *δ*_H_ 1.98 (d, *J* = 1.8 Hz, H-12 or H-13), *δ*_H_ 1.80 (s, H-12 or H-13), *δ*_H_ 1.24 (s, H-14), and *δ*_H_ 1.77 (s, H-15).

The ^13^C spectrum showed *δ*_C_ 202.7 signal for carbonyl carbon. There were four methyl carbons at *δ*_C_ 23.2 (C-12 or C-13), *δ*_C_ 22.3 (C-12 or C-13), *δ*_C_ 31.1 (C-14), and *δ*_C_ 21.6 (C-15) and four methynic carbons at *δ*_C_ 110.4 (C-1), *δ*_C_ 140.6 (C-2), *δ*_C_ 137.8 (C-3) and *δ*_C_ 50.9 (C-5). With the analysis of the HMBC spectrum, it was possible to observe coupling via three bonds (^3^*J*_CH_) of C-3 (*δ*_C_ 137.8) with H-5 (*δ*_H_ 2.2) and H-15 (*δ*_H_ 1.8) and coupling via two bonds (^3^*J*_CH_) of C -2 (*δ*_C_ 140.6) with H-1 (*δ*_H_ 4.4) and H-3 (*δ*_H_ 6.1). These couplings infer the position of C-2 and C-3, which showed different displacements to those reported in the literature, C-2 (*δ*_C_ 137.7) and C-3 (*δ*_C_ 140.5) [8].

Compounds **1** and **2** were evaluated for their antioxidant potential using the DPPH assay, and the results are shown in Figure 2. It is noteworthy that compounds **1** and **2** present insignificant free-radical scavenging activity. Regarding the inhibition of lipoperoxidation in rat brain homogenate, the compounds **1** and **2** did not show inhibitory activity at 40 mg/L. Thus, it is clear that compounds **1** and **2** do not present significant antioxidant activity through either single-electron transfer or hydrogen atom transfer (HAT) mechanisms. Garmus (2014) evaluated the antioxidant activity of the essential oil extracted from *E. uniflora* leaves and found that phenolic compounds found in the oil are mainly responsible for the antioxidant effect (determined using the DPPH assay). Similarly, Auricchio (2007) studied a hydroalcoholic extract from *E. uniflora* leaves and its inhibition of lipid oxidation in rat brain homogenate and found an IC_50_ of 35 mg/L, which is different from the results obtained herein. Considering the data are obtained using two distinct mechanisms of antioxidant action, it is hypothesized that compounds **1** and **2** have no antioxidant potential [20,21].

Regarding the cytotoxicity assay, the parameters IC_50_, GI_50_, and LC_50_ correspond to cell viability, antiproliferative effect, and cell death, respectively (Figure 3). All cell lines presented IC_50_ values > 500 µM, except **2** for HepG2 cells (IC_50_ = 525.4 µM). Following a similar approach, Figueiredo et al. (2019) reported that *E. uniflora* oil, with significant amounts of **1** and **2,** decreased cell viability (IC_50_ = 15.42 µg/mL) of MRC-5 normal cells [12]. Compounds **1** and **2** promoted, most notably, antiproliferative effects on IMR90 normal cells (GI_50_ = 184.7 and 14.9 µM, respectively) when compared with A549 (GI_50_ = 590.8 and 359.3 µM, respectively) and HepG2 (GI_50_ = 289.2 and 97.5 µM, respectively) cancer cells, indicating high cytotoxicity and low safety in in vitro studies. The lethal concentration (LC_50_) for **1** and **2** was higher than 500 µM, indicating that it is necessary to use higher concentrations of these compounds to kill half of the cells. In the literature, it is recognized that phenolic compounds can act as either anti- or pro-oxidant agents [22] depending on their concentration, environmental pH, and the presence of metals and oxygen [23].

Herein, despite their antioxidant effect as pointed out by the DPPH assay, it was clear that both **1** and **2** compounds exerted pro-oxidant behavior (Figure 4) by inducing reactive oxygen species (ROS) generation in non-cancer (IMR90) and malignant (A549) cells, which explains their cytotoxicity observed in cell viability assay. It is known that if ROS levels increase dramatically to toxic concentrations, the JNK (c-Jun NH2-terminal kinases) pathway can be activated, resulting in apoptosis and cell death [22]. By contrast, Sobeh (2019) reported that the methanolic extract of *E. uniflora* leaves present noticeable antioxidant properties by reducing the intracellular ROS levels and by increasing the reduced glutathione (GSH) content in HaCaT cells [24]. This disagreement may be explained by the chemical profile of a crude extract compared to the isolated compounds **1** and **2**. In this case, the antioxidant effect may have occurred due to the synergistic effects between the bioactive compounds the crude extract, while the oxidative activity shown herein was related to the individual abilities of compounds **1** and **2**. Indeed, our results highlight the high cytotoxicity of compounds **1** and **2**, acting in a more intense way against IMR90 normal cells than cancer cells (A549 and HepG2) because of their pro-oxidant behavior, as observed in the ROS generation assay.

The antioxidant activity measured by chemical and biological assays, together with the cell-based antioxidant activity measurement, clearly indicates that **1** and **2** cannot be considered antioxidant agents. Data obtained using chemical antioxidant activity (inhibition of lipid peroxidation and DPPH assay) and the induced ROS generation show that **1** and **2** are not able to scavenge free radicals (DPPH, peroxyl and hydroxyl radicals) and decrease intracellular ROS generation. Thus, it is clear that any prospect of compounds **1** and **2** having antioxidant action is very limited. In addition, taken all together, results show pro-oxidative effects and cytotoxicity in relation to normal human cells. This poses a toxicological concern for both substances, and the data generated here can be used as a basis for other biological assays that may encompass antioxidant and cytotoxic studies of the compounds [25].

## 3. Materials and Methods

### 3.1. Plant Material

*Eugenia uniflora* L. (Myrtaceae) leaves were collected in the Uvaranas Campus, Ponta Grossa State University, Ponta Grossa—PR, Brazil, in the location −25.091824 and −50.102573, and an exsiccate was deposited at the HUPG (Herbarium University of Ponta Grossa) under number 22452.

### 3.2. Essential Oil Extraction and GC–MS Analysis

Three portions of approximately 300 g of ground dried leaves were extracted for 2.5 h in a steam distillation glass distiller. The oil samples were pooled in ethyl ether, dried over anhydrous Na_2_SO_4_, filtered, and evaporated under vacuum and at low temperature, yielding 7.56 g (0.84% *w*/*w*) based on dry weight.

For the oil analyses by GC–MS (Shimadzu GCMS-QP2010 Plus Gas Chromatograph), a non-polar Rtx-5MS column (30 m × 0.25 mm × 0.25 μm) and the following analytical conditions were used: split ratio of 1/20, 250 °C for the injector, 250 °C for the ion source, and 280 °C for the interface. The oven temperature was programmed to 60 °C for the first 5 min, increasing at a rate of 3 °C/min to reach the final temperature of 240 °C. The components were identified based on the relative retention indices calculated using a series of n-alkanes (C8–C19) and the mass spectra from the apparatus database, followed by comparison with the published data [8,17,18].

### 3.3. Isolation and Structure Identification of Compounds

The essential oil sample (7.0 g) was fractionated on a hexane packed silica gel 60 (240–400 mesh, Merck) column chromatography eluted with hexane/ethyl acetate mixtures in a gradient of increasing polarity. The two main components of the oil were isolated in various fractions (**1**, 0.6 g; **2**, 0.4 g) and presented as single well-defined spots in thin layer chromatography (Macharey-Nagel 60 G_254_ 0.2 mm plates) first observed under 254 nm UV light and then sprayed with H_2_SO_4_/MeOH (1:1) and heated on a hot plate.

Structural identification of the compounds was carried out by spectroscopic analysis. Optical rotation measurements were determined on a Perkin-Elmer 343 polarimeter. UV spectra (in CHCl_3_) were measured on a Varian Cary 50 spectrophotometer. ^1^H and ^13^C NMR spectra were obtained on a Bruker Ascend 400 MHz spectrometer using CDCl_3_ and tetramethylsilane as internal reference. Signal assignments in NMR spectra were made by combining the 1D and 2D techniques, correlation spectroscopy (COSY), heteronuclear single quantum correlation (HSQC), and heteronuclear multiple bond correlation (HMBC) and by comparison with data published in the literature.

**1**: Oil; ${\left[\alpha \right]}_{D}^{20}-8{}^{\circ}$ (*c* 1.0, CHCl_3_); UV (CHCl_3_) λ_max_: 250 nm; ^1^H NMR (400 MHz, CDCl_3_), see Table 2; ^13^C NMR (100 MHz, CDCl_3_), see Table 2; HMBC (selected correlations) C1→H3, H9/H9’, H14; C2→H15; C3→H2, H1, H15; C4→H15, H2; C5→H3, H6; C6→H5, H15; C7→H13, H12; C10→H14, H2, H1; C14→H9/H9’; C15→H3.

**2**: Oil; ${\left[\alpha \right]}_{D}^{20}-144{}^{\circ}$ (*c* 1.0, CHCl_3_); UV (CHCl_3_) λ_max_: 250 nm; ^1^H NMR (400 MHz, CDCl_3_), see Table 2; ^13^C NMR (100 MHz, CDCl_3_), see Table 2; HMBC (selected correlations) C1→H2, H9/H9’, H14; C2→H3, H1; C3→H15, H5, H2; C4→H3, H15; C5→H3, H6; C7→H12, H13; C10→H14, H2; C14→H9/H9’; C15→H3.

### 3.4. Chemical and Biological Antioxidant Activities

The free-radical scavenging activity of **1** and **2** in relation to the 2,2-diphenyl-1-picrylhydrazyl (DPPH) radical was measured using the colorimetric method described by Brand-Williams (1995), and the results are expressed in mg of ascorbic acid equivalent per 100 g of material (AAE/100 g) [26]. To evaluate the capacity of hydrogen atom transfer (HAT) of the isolated compounds, male Wistar rat brain homogenate was used as substrate for the production of thiobarbituric acid reactive substances (TBARS) produced by Fe^2+^-induced oxidation undertaken at 37 °C, following the experimental conditions described elsewhere [27]. The inhibition of lipid peroxidation was expressed as % of inhibition. The animal protocol was approved by the Ethics Committee (UEPG, protocol 47/2017).

### 3.5. Cytotoxicity Assay

The in vitro cytotoxic effect of **1** and **2** compounds were analyzed in relation to A549 (lung adenocarcinoma epithelial cells—BCRJ code: 0033), HepG2 (human hepatoma carcinoma cells—BCRJ code: 0291), and IMR90 (human lung fibroblast—BCRJ code: 0118) cell lines using MTT (3-(4,5-dimethylthiazol-2-yl)-2,5-diphenyl tetrazolium bromide) (Sigma-Aldrich, COD—M5655) assay [28]. All cell lines were maintained in Dulbecco’s Modified Eagle’s Medium/Nutrient Mixture F-12 Ham (Sigma-Aldrich, COD—D8900), with added 10% fetal bovine serum (FBS, Gibco, COD—16000044) and 100 IU penicillin/100 µg streptomycin (Sigma-Aldrich, COD—P4333). The cell lines were grown in a humidified incubator at 37 °C containing 5% CO_2_. Briefly, the cells were seeded into 96-well plates at a density of 1 × 10^4^ cells/well (HepG2 and A549), 5 × 10^3^ cells/well (IMR90), 100 μL/well, and after 24 h, the cells were treated with 100 μL of serial concentrations (10–500 μM) of compounds **1** and **2** for 48 h. The stock solution of extracts contained 5% of dimethyl sulfoxide (DMSO) at a final concentration of 0.2% in cell medium for all in vitro assays. The IC_50_, GI_50_, and LC_50_ parameters were performed in accordance with the method described by do Carmo (2018) [22], in which IC_50_ is the concentration of the agent that inhibits growth by 50%, calculated as (T/C) × 100 = 50, where T = number of cells at time t of treatment; C = control cells at time t of treatment; GI_50_ is the concentration of the agent that inhibits growth by 50% relative to untreated cells, calculated as ((T − T0)/(C − T0)) × 100 = 50, where T and C are the number of treated and control cells, respectively, at time t of treatment, and T > T0 where T0 is the number of cells at time zero; LC_50_ is the concentration of the agent that results in a net loss of 50% cells relative to the number at the start of treatment, calculated as ((T − T0)/T0) × 100 = −50, where T < T0. The experiments were conducted in quadruplicate on three different days to assure the reproducibility of the generated data.

### 3.6. Intracellular Reactive Oxygen Species (ROS) Measurement

In order to assess the in vitro antioxidant/oxidant potential of **1** and **2**, intracellular ROS generation was measured using a fluorescence technique as described by Carmo (2019) [29]. Briefly, A549 cancer cells (6 × 10^4^/well) and IMR90 normal cells (2 × 10^4^/well) were exposed to concentrations of 10, 50, and 100 μg/mL of **1** and **2** compounds, culture medium (negative control), and 15 μmol/L H_2_O_2_ (positive control), for 1 h in DCFH-DA solution (25 mmol/L), at 37 °C and 5% CO_2_. Subsequently, cells were washed with PBS and a H_2_O_2_ solution (15 μmol/L) added for recording of fluorescence. Intracellular fluorescence intensity of cells was measured at an excitation wavelength of 485 nm and at an emission wavelength of 538 nm [29]. The assays were performed in quadruplicate on three different days.

### 3.7. Statistical Analysis

The significance for intracellular ROS measurement was defined by one-way analysis of variance followed by Tukey’s test. Analysis on sigmoidal dose–response for cytotoxicity was performed using nonlinear regression for curve fitting.

## 4. Conclusions

We obtained and confirmed the chemical structures of **1** and **2** here and also provide a simple method for their isolation. Our results for structure analyses and cytotoxicity confirm the importance of establishing more knowledge about the volatile metabolites of *Eugenia uniflora*.

## Figures and Tables

**Figure 1 molecules-26-00740-f001:**
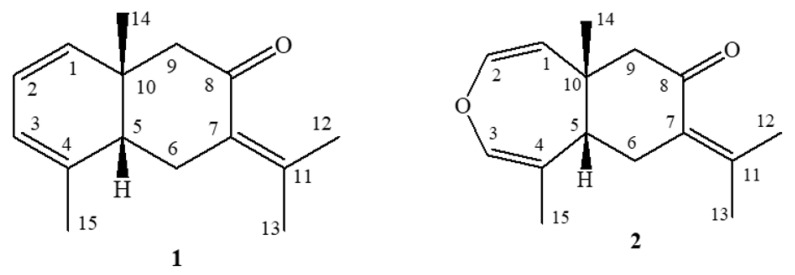
Chemical structures of selina-1,3,7(11)-trien-8-one (**1**) and oxidoselina-1,3,7-trien-8-one (**2**) presenting the relative stereochemistry in C-5 and C-10, as proposed in the literature [8].

**Figure 2 molecules-26-00740-f002:**
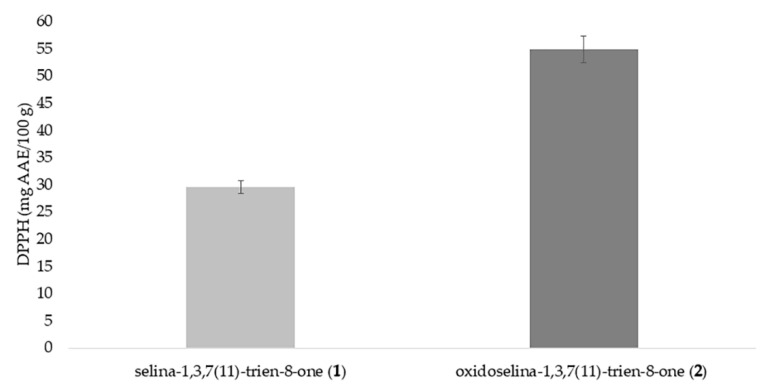
Antioxidant activity of compounds **1** and **2** in mg of ascorbic acid equivalent (AAE) per 100 g of material (AAE/100 g) as measured by the DPPH assay.

**Figure 3 molecules-26-00740-f003:**
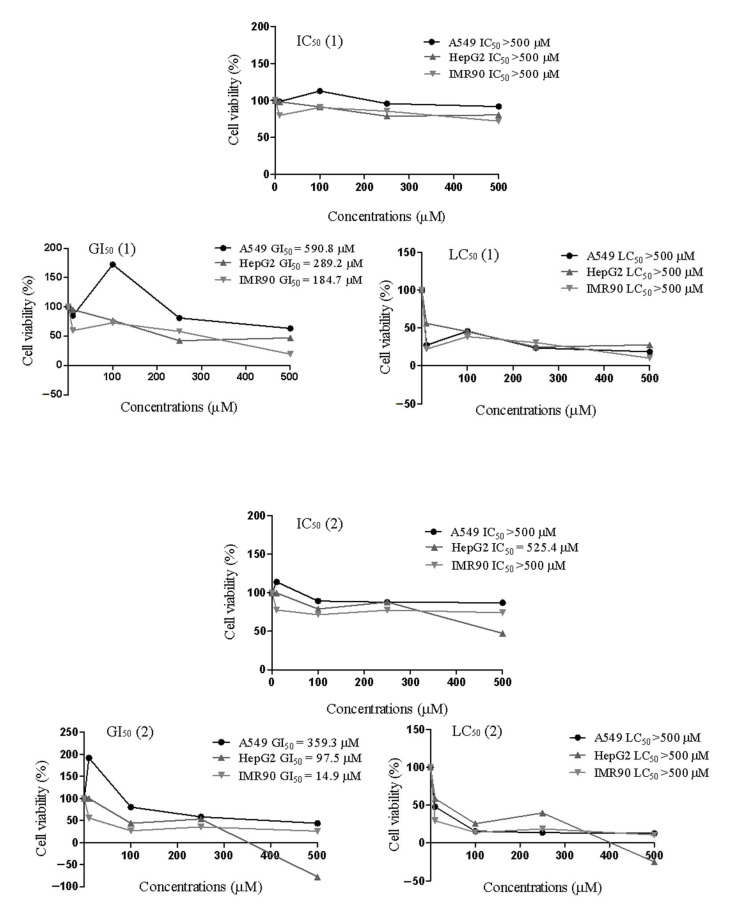
Cell viability of human lung adenocarcinoma epithelial (A549), human hepatoma carcinoma (HepG2), and normal human lung fibroblast (IMR90) cells treated with compounds **1** and **2**.

**Figure 4 molecules-26-00740-f004:**
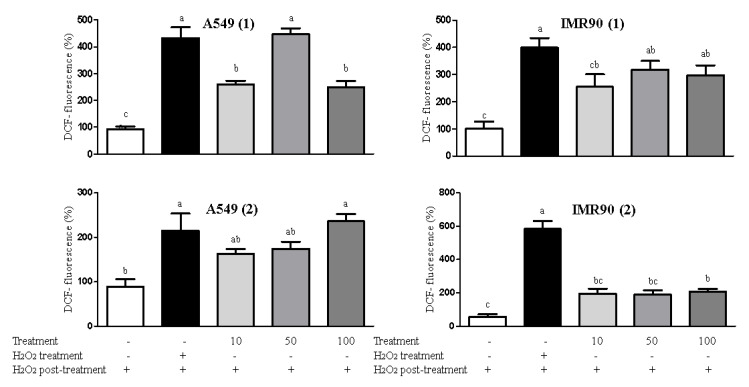
Results for intracellular ROS in A549 and IMR90 cells treated with **1** and **2** (10, 50 and 100 μM) as measured by spectrofluorometry. The data were analyzed by one-way ANOVA and quantitative data are expressed by mean ± standard deviation. * Different letters in the columns mean statistical difference.

**Table 1 molecules-26-00740-t001:** Relative composition (%) of the identified components in the GC–MS analysis of the essential oil from leaves of *E. uniflora*.

R_t_ (min)	%	RRI ^a^	RRI ^b^	Compound Name
28.015	3.60	1392.8	1392	β-elemene
29.174	0.98	1419.9	1420	caryophyllene
29.760	0.83	1434.7	1434	γ-elemene
31.942	0.11	1482.5	1480	germacrene D
32.306	0.81	1493.8	-	M + 204
32.404	4.72	1499.2	1500	curzerene
34.764	1.74	1562.5	1561	germacrene B
35.596	9.27	1580.3	1578	spathulenol
35.809	6.66	1586.4	1588	epiglobulol
36.131	1.22	1594.5	-	M + 222
37.639	13.34	1632.1	1631	selina-3,5,7(11)-trien-8-one (**1**)
38.505	2.62	1658.5	1658	selina-6-en-4-ol
39.661	1.27	1667.9	1666	furanodiene
40.025	2.92	1696.6	1693	germacrone
40.031	4.62	1700.1	-	M + 232
40.824	3.58	1722.1	-	M + 220
41.821	20.44	1755.8	1757	oxidoselina-1,3,7(11)-trien-8-one (**2**)
42.505	1.92	1769.9	-	M + 220
42.612	2.20	1772.4	-	M + 220
43.509	1.08	1797.5	-	M + 218

RRI ^a^: values of calculated relative retention indices using the column Rtx-5MS (GC–MS) and the n-alkanes series C8–C19; RRI ^b^: published relative retention indices for apolar columns [17,18].

**Table 2 molecules-26-00740-t002:** ^1^H and ^13^C NMR spectra data for **1** and **2** isolated from the essential oil of *E. uniflora*.

H/C	1		2	
	*δ*_H_ (*J* Hz)	*δ* _C_	*δ*_H_ (*J* Hz)	*δ* _C_
1	5.63 (d, 5.2)	131.7	4.41 (d, 6.5)	110.4
2	5.76 (dd, 9.4, 5.3)	123	6.02 (d, 7.7)	140.6
3	5.33 (d, 9.4)	118	6.13 (s)	137.8
4	-	138.3	-	119.1
5	2.00 (dd, 10.6, 4.8)	46.1	2.21 (dd, 12.9, 6.3)	50.9
6	2.66 (dd, 10.6, 4.8)	29.8	2.83 (dd, 15.4, 4.8)	32.9
6’	2.24 (m)	29.8	2.44–2.54 (m)	32.9
7	-	132.6	-	130.7
8	-	203.9	-	202.7
9	2.47 (d, 14.5)	53.4	2.40 (d, 15.0)	57.8
9’	2.30 (d, 14.5)	53.4	2.19 (d, 13.7)	57.8
10	-	38.5	-	41
11	-	139.6	-	143.6
12 or 13	1.79 (s)	21.7	1.80 (s)	22.3
12 or 13	1.94 (d, 1.6)	22.6	1.98 (d, 1.8)	23.2
14	1.04 (s)	26.7	1.24 (s)	31.1
15	1.85 (s)	22.2	1.77 (s)	21.6

NMR assignments are based on ^1^H–^13^C HSQC, HMBC, and ^1^H–^1^H COSY contour maps; s: singlet; m: multiplet; d: doublet; dd: double doublets.

## Data Availability

Data is contained within the article or Appendix A.

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
