# Peer review of "Selina-1,3,7(11)-trien-8-one and Oxidoselina-1,3,7(11)-trien-8-one from Eugenia uniflora Leaf Essential Oil and Their Cytotoxic Effects on Human Cell Lines"

_molecules, 2021, doi:10.3390/molecules26030740_

Round 1

Reviewer 1 Report

Dear Authors,

  1. The manuscript requires grammar and style correction. English language and style should be revised, preferably by a proof-reader with good written English skills.
  2. The manuscript lacks the description of materials used in experiments. Please at least give the types and manufacturers of cell media, FBS, PBS, antibiotics and MTT. What concentration of FBS in cell media was used for experiments?
  3. Please give appropriate catalogue numbers of the cell lines.
  4. Please provide conditions of cell line incubation
  5. There is no information on the preparation of stock solutions of extracts for in-vitro studies. If organic solvents as DMSO or ethanol were used their influence on those studies should be tested. What was the solvent used for preparation of stock solutions?. What was the concentration of stock solutions? What was the highest concentration of solvent in cell media? Was the cytotoxicity of solvent present in dilutions of stock solution and the influence of solvent on other in vitro studies tested?
  6. How many times were the experiments repeated? What was the number of replicates?
  7. Please explain what IC50, GI50 and LC50 values are. You have tested so many parameters without describing and discussing the obtained results. In the results and discussion you only mention GI50. In the following discussion you only mention IC50 provided by cited references which is completely different parameter than GI50. Please explain.
  8. Figure 3 lacks descriptions. What do the letters A, B, C, D, E, F, G refer to? Results presented in Figure 3 are extremely poorly described in the text.
  9. Line 26-26 – you write “Both compounds did not show prominent free-scavenging activity in relation to DPPH” and then line 202 – “despite their antioxidant effect pointed out by DPPH assay”, whereas further in line 217-218 - “1 and 2 cannot be considered antioxidant agents neither using chemical reactions nor using cell-based experiments”. Please decide whether your compounds are antioxidant or not.
  10. Line 173 – you have used 40 mg/mL in the inhibition of lipoperoxidation of rat’s brain homogenate test. What was the reason for selecting so high concentration? Auricchio et al. observed the 50% inhibition at 34,6 μg/mL.

Author Response

Dear Reviewer 1.
We would like to thank the reviewer for his contributions to our manuscript. We readily accepted all suggestions and made changes as recommended. Find below and in the attached manuscript all the answers to the suggestions.
Best regards.

  1. The manuscript requires grammar and style correction. English language and style should be revised, preferably by a proof-reader with good written English skills.
    Response 1: Thanks for the suggestion. The entire text has been revised for English grammar and the style of Molecules magazine.

  1. The manuscript lacks the description of materials used in experiments. Please at least give the types and manufacturers of cell media, FBS, PBS, antibiotics and MTT. What concentration of FBS in cell media was used for experiments?
    Response 2: Thanks for the suggestion. The description of materials were added in 3.5 item. According to American Type Culture Collection (ATCC) recommendations, we used 10% of FBS in cell media.

  1. Please give appropriate catalogue numbers of the cell lines.
    Response 3: Thanks for the observation. We provided the catalogue numbers of the cell lines. Please see lines 297 -299).

  1. Please provide conditions of cell line incubation
    Response 4: The conditions of cell line incubation were added in item 3.5 (see lines 294 – 317):

“All cell lines were maintained in Dulbecco’s Modified Eagle’s Medium/Nutrient Mixture F-12 Ham (Sigma-Aldrich, COD – D8900), with added 10% fetal bovine serum (FBS, Gibco, COD - 16000044) and 100 IU penicillin/100 µg streptomycin (Sigma-Aldrich, COD – P4333). The cell lines were grown in a humidified incubator at 37 °C containing 5% CO2.”

  1. There is no information on the preparation of stock solutions of extracts for in-vitro studies. If organic solvents as DMSO or ethanol were used their influence on those studies should be tested. What was the solvent used for preparation of stock solutions?. What was the concentration of stock solutions? What was the highest concentration of solvent in cell media? Was the cytotoxicity of solvent present in dilutions of stock solution and the influence of solvent on other in vitro studies tested?
    Response 5: Thanks for the observation. We provided this information in the text in item 3.5 (see lines 294 – 317).

The stock solution of extracts was prepared with 95% water and 5% of DMSO, in order to dilute the sample. The final concentration of DMSO in cell media was 0.2% for all treatments, including the control. According to Qi, Ding & Salvi (2008) [1], DMSO concentrations of 0.1% and 0.25% in cell cultures, caused no cytotoxic effects. Thus, it is expected that the final concentration of DMSO used in our experiments did not cause cell damage.

  1. Qi, W.; Ding, D.; Salvi, R.J. Cytotoxic effects of dimethyl sulphoxide (DMSO) on cochlear organotypic cultures. Hear. Res. 2008, 236, 52–60, doi:10.1016/j.heares.2007.12.002.

  1. How many times were the experiments repeated? What was the number of replicates?
    Response 6: Thanks for the contribution. The experiments were conducted in quadruplicate on three different days, to assure the reproducibility of the generated data. This information was inserted in the manuscript in item 3.5 (see lines 315 -317).

  1. Please explain what IC50, GI50and LC50 values are. You have tested so many parameters without describing and discussing the obtained results. In the results and discussion you only mention GI50. In the following discussion you only mention IC50 provided by cited references which is completely different parameter than GI50. Please explain.
    Response 7: We agree with the reviewer. We explained the meaning of IC50, GI50 and LC50 parameters, improved the discussion and adjusted the terms for each parameter properly in item 3.5 (see lines 307 -315). Thanks for the collaboration.

  1. Figure 3 lacks descriptions. What do the letters A, B, C, D, E, F, G refer to? Results presented in Figure 3 are extremely poorly described in the text.
    Response 8: Thanks for the observation. Sorry our mistake, we forgot to remove these letters from the Figure 3, now it is correct. Moreover, the results are better explored in the text.

  1. Line 26-26 – you write “Both compounds did not show prominent free-scavenging activity in relation to DPPH” and then line 202 – “despite their antioxidant effect pointed out by DPPH assay”, whereas further in line 217-218 - “1 and 2 cannot be considered antioxidant agents neither using chemical reactions nor using cell-based experiments”. Please decide whether your compounds are antioxidant or not.
    Response 9: Thanks for the observation. Neither the compounds have antioxidant potential. Changes were made in the revised version.

  1. Line 173 – you have used 40 mg/mL in the inhibition of lipoperoxidation of rat’s brain homogenate test. What was the reason for selecting so high concentration? Auricchio et al. observed the 50% inhibition at 34,6 μg/mL.
    Response 10: Thanks for the observation. Using a biological medium to assess antioxidant potential is an interesting approach. However, in our previous research, quercetin is a good inhibitor of lipid oxidation of Wistar rat’s brain homogenate. Quercetin is usually used from 15 to 50 mg/L, which is not high at all. Over 100 mg/L may be considered high. Overall, there is not a problem

References:

GREMSKI, L. M., COELHO, L. K., SANTOS, J. S., DAGUER, H., MOLOGNONI, L., PRADO-SILVA, L., SANT’ANA, A. S., ROCHA, R. S., DA SILVA, M. C., CRUZ, A. G., AZEVEDO, L., DO CARMO, M. A. V., WEN, M., GRANATO. D. Antioxidants-rich ice cream containing herbal extracts and futooligossaccharides: manufacture, functional and sensory properties. Food Chemistry, v. 298, ID: 125098, 2019.

ESCHER, G. B., BORGES, L. C., SANTOS, J. S., CRUZ, T. M., MARQUES, M. B., do CARMO, M. A. V., AZEVEDO, L., FURTADO, M. M., SANT’ANA, A. S., WEN, M., ZHANG, L., GRANATO, D. From the field to the pot: phytochemical and functional analyses of Calendula officinalis L. flower for incorporation in an organic yogurt. Antioxidants, v. 8. n.11, p. 559-582, 2019.

Reviewer 2 Report

The manuscript “Selina-1,3,7(11)-trien-8-one and Oxidoselina-1,3,7(11)-trien-8-one from Eugenia uniflora leaf essential oil and their cytotoxic effects on human cells lines” describes silica-gel column chromatography isolation and purification of Selina-1,3,7(11)-trien-8-one and Oxidoselina-1,3,7(11)-trien-8-one from the leaves of Eugenia uniflora. The antioxidant and cytotoxic activities were evaluated for these two compounds. The work was carefully conducted and deserves to be published after minor corrections. I believe this work is fit for publication, but I have some minor corrections and suggestions that I suggest the authors to consider (see below).

Most important: The original FID of the NMR spectra (including COSY, HSQC and HMBC) must be provided by the authors in the Supplementary Material.

Line 159: 2JCH

Suggestion: Coupling of C-2 with H-3 is 3JCH.

Line 175-179: Antioxidant activities against the DPPH radical reported by Garmus (2014), and the inhibitory capacity of lipoperoxidation in rat brain homogenate identified by Auricchio (2007) in the essential oil of E. uniflora leaves are not directly related to the compounds 1 and 2 isolates and possibly other compounds present in the leaves and/or synergistic effects between more than one compound.

Suggestion: This sentence does not read correctly. Please amend.

Figure 1.

Suggestion: Include the NMR spectra in Figure 1 and the structures could go on top of the NMR spectra.

Oxidoselina-1,3,7(11)-trien-8-one

InChI=1S/C15H20O2/c1-10(2)12-7-13-11(3)9-17-6-5-15(13,4)8-14(12)16/h5-6,9,13H,7-8H2,1-4H3

Smiles: CC12C(C/C(C(C2)=O)=C(C)\C)C(C)=COC=C1

https://webbook.nist.gov/

Oxidoselina-1,3,7(11)-trien-8-one

InChI=1S/C15H20O2/c1-10-6-5-7-14(4)9-12(16)15(8-11(10)14)13(2,3)17-15/h5-7,11H,8-9H2,1-4H3/t11?,14-,15?/m0/s1

Smiles

CC(C1([H])CC23C(C)(O3)C)=CC=C[C@]1(CC2=O)C

Suggestion:

How about using one of these names alongside Oxidoselina-1,3,7(11)-trien-8-one?

rel-(5aR,9aR)-3-Benzoxepin-7(6H)-one,5a,8,9,9a-tetrahydro-1,5a-dimethyl-8-(1-methylethylidene)

Or

rel-(5aR,9aR)-1,5a-dimethyl-8-(propan-2-ylidene)-5a,8,9,9a-tetrahydrobenzo[d]oxepin-7(6H)-one

Lines 124-128: Compound 1 was isolated in our laboratory as an oil and presented optical activity [α] 20D = -8° 124 (c 1.0, CHCl3), very close to the value registered in the literature when the compound was isolated for the first time [8]. Compound 1 synthesized by Kanazawa and colleagues [24] in 2000 presented the value of [α] 20D = -258° (c 1.0, CHCl3), which can be now considered as a very discrepant value when compared to that found for the natural substance.

Suggestion: The NMR spectra presented in the Supplementary Material for compounds 1 and 2 have at least 20% impurity. Therefore, this kind of comparison and statement should not be done.

Line 320: … as described Carmo (2019).

Suggestion: Use the numbering citation instead of name citation.

Author Response

Dear Reviewer 2.
We would like to thank the reviewer for his contributions to our manuscript. We readily accepted all suggestions and made changes as recommended. Find below and in the attached manuscript all the answers to the suggestions.
Best regards.

  1. The original FID of the NMR spectra (including COSY, HSQC and HMBC) must be provided by the authors in the Supplementary Material.
    Response 1: Thanks for the suggestion. We added the COZY, HSQC and HMBC spectra in the supplementary material as suggested. I will e-mail the editor because there is no adding two files here (manuscript and supplementary material corrected).
  1. Line 159: 2JCH Suggestion: Coupling of C-2 with H-3 is 3JCH.
    Response 2: Thanks for the suggestion, see line 147.
  1. Line 175-179: Antioxidant activities against the DPPH radical reported by Garmus (2014), and the inhibitory capacity of lipoperoxidation in rat brain homogenate identified by Auricchio (2007) in the essential oil of E. uniflora leaves are not directly related to the compounds 1 and 2 isolates and possibly other compounds present in the leaves and/or synergistic effects between more than one compound. Suggestion: This sentence does not read correctly. Please amend.
    Response 3: Thanks for the suggestion. Changes were made in the revised version.
  1. Figure 1. Suggestion: Include the NMR spectra in Figure 1 and the structures could go on top of the NMR spectra.
    Response 4: Thanks for the suggestion, but note that the location of the figure at this point is good for presenting the structures and numbers, so that the reading of the text can be well monitored, so we prefer to leave the figures of spectra for the Material Supplementary.
  1. Oxidoselina-1,3,7(11)-trien-8-one. InChI=1S/C15H20O2/c1-10(2)12-7-13-11(3)9-17-6-5-15(13,4)8-14(12)16/h5-6,9,13H,7-8H2,1-4H3. Smiles: CC12C(C/C(C(C2)=O)=C(C)\C)C(C)=COC=C1
    https://webbook.nist.gov/
    Oxidoselina-1,3,7(11)-trien-8-one
    InChI=1S/C15H20O2/c1-10-6-5-7-14(4)9-12(16)15(8-11(10)14)13(2,3)17-15/h5-7,11H,8-9H2,1-4H3/t11?,14-,15?/m0/s1
    Smiles: CC(C1([H])CC23C(C)(O3)C)=CC=C[C@]1(CC2=O)C
    Suggestion: How about using one of these names alongside Oxidoselina-1,3,7(11)-trien-8-one? rel-(5aR,9aR)-3-Benzoxepin-7(6H)-one,5a,8,9,9a-tetrahydro-1,5a-dimethyl-8-(1-methylethylidene) Or rel-(5aR,9aR)-1,5a-dimethyl-8-(propan-2-ylidene)-5a,8,9,9a-tetrahydrobenzo[d]oxepin-7(6H)-one
    Response 5: Thanks for the suggestion. We add to the text, at the first point where the names of both compounds 1 and 2 appear, their names referring to the relative settings as they were originally suggested by Weyrstahl et al (1988) reference 8.

  1. Lines 124-128: Compound 1 was isolated in our laboratory as an oil and presented optical activity [α] 20D = -8° 124 (c 1.0, CHCl3), very close to the value registered in the literature when the compound was isolated for the first time [8]. Compound 1 synthesized by Kanazawa and colleagues [24] in 2000 presented the value of [α] 20D = -258° (c 1.0, CHCl3), which can be now considered as a very discrepant value when compared to that found for the natural substance. Suggestion: The NMR spectra presented in the Supplementary Material for compounds 1 and 2 have at least 20% impurity. Therefore, this kind of comparison and statement should not be done.
    Response 6: Very grateful for your suggestion, however, compound 1 was isolated here from the same plant source and presented a very similar specific optical rotation. As we know from practice, a small value for the specific optical rotation requires a larger sample with high purity, requirements without which the value that indicates the same stereochemistry for the two isolates of the natural compound would not be obtained. In addition, our analyzes by several NMR experiments presented and discussed in the text also showed significant differences in relation to the assignments made by Kanazawa et al (2000), reinforcing the suggestion that the synthetic compound has a different stereochemistry than the natural compound 1.
  1. Line 320: … as described Carmo (2019). Suggestion: Use the numbering citation instead of name citation.
    Response 7: Thanks for the suggestion. See line 322.

Reviewer 3 Report

Line 46: E. uniflora must be written in italics 

Line 83-84: E. uniflora must be written in italics 

Line 120: why the reference (Kanazawa) is labeled as 24 instead of 19?

Line 199: Authors can also express IC50 value in microM (as GI50).

Line 219: What is ILP? Inhibitory capacity of lipoperoxidation?

Why did authors choose 40 mg/mL of both compounds for measuring HAT activity? And how much for DPPH?  Later they talk in microM so, express in same units. 

Figure 3: Where are control cells?

Figure 3: Can you explain results from E?

Figure 3: Attending to C and F graphs, Is not LC50 less than 500 micM?

Figure 4: Did authors not test on HepG2?

Figure 4: Any significant difference? What are the meaning of the letters?

Line 314: Why 48 h? 

Line 359: Check reference style

Line 370: Check reference

Why did authors select those human cell lines?

Authors must improve introduction section (meaning of essential oils, sesquiterpenes, human cancer cells...)

If they can perform some other antioxidant assay would be nice... (FRAP assay, superoxide radical...)

Author Response

Dear Reviewer 3.
We would like to thank the reviewer for his contributions to our manuscript. We readily accepted all suggestions and made changes as recommended. Find below and in the attached manuscript all the answers to the suggestions.
Best regards.

  1. Line 46: uniflora must be written in italics
    Response 1: Thanks for the correction. See line 47.
  1. Line 83-84: uniflora must be written in italics
    Response 2: Thanks for the correction. See line 86.
  1. Line 120: why the reference (Kanazawa) is labeled as 24 instead of 19?
    Response 3: Thanks for the observation. Sorry our mistake, changes were made in the revised adjusting the position of references 19 – 29.
  1. Line 199: Authors can also express IC50 value in microM (as GI50).
    Response 4: Thanks for the suggestion. The authors used oil y extract with complex matrix with different compounds in its composition. For expressing the IC50 value in µM, it would be necessary the molar mass data. Herein, our compound is pure and the molar mass is known.
  1. Line 219: What is ILP? Inhibitory capacity of lipoperoxidation?
    Response 5: ILP = inhibition of lipid peroxidation. Changes were made in the revised version.
  1. Why did authors choose 40 mg/mL of both compounds for measuring HAT activity? And how much for DPPH? Later they talk in microM so, express in same units.
    Response 6: Thanks for the question. In fact, there was a mistake in the unit.  We used a concentration of 40 mg/L. In our previous experiments, a concentration between 15 to 50 mg/L of positive control (quercetin) was used. Thus, as we are using isolated compounds, it is not appropriate to use concentrations higher than 50 mg/L or any potential benefitial effects would be inflated.

References:

GREMSKI, L. M., COELHO, L. K., SANTOS, J. S., DAGUER, H., MOLOGNONI, L., PRADO-SILVA, L., SANT’ANA, A. S., ROCHA, R. S., DA SILVA, M. C., CRUZ, A. G., AZEVEDO, L., DO CARMO, M. A. V., WEN, M., GRANATO. D. Antioxidants-rich ice cream containing herbal extracts and futooligossaccharides: manufacture, functional and sensory properties. Food Chemistry, v. 298, ID: 125098, 2019.

ESCHER, G. B., BORGES, L. C., SANTOS, J. S., CRUZ, T. M., MARQUES, M. B., do CARMO, M. A. V., AZEVEDO, L., FURTADO, M. M., SANT’ANA, A. S., WEN, M., ZHANG, L., GRANATO, D. From the field to the pot: phytochemical and functional analyses of Calendula officinalis L. flower for incorporation in an organic yogurt. Antioxidants, v. 8. n.11, p. 559-582, 2019.

  1. Figure 3: Where are control cells?
    Response 7: Thanks for the question. The control cells are represented at concentration of 0 µg/mL (without treatment) and cell viability of 100% for all parameters and cell lines. Observe that all of them started at the same point in the graphs (100% of cell viability and 0 µg/mL).
  1. Figure 3: Can you explain results from E?
    Response 8: The compound 2 exerted antiproliferative effects for all cell lines. For A549, low its concentrations stimulated cell proliferation, however, when the concentration increased this compound promoted reduction on cell proliferation (GI50 = 359.3 µM), until stabilization. Similarly, for IMR90 normal cells, this antiproliferative effect stabilized after its GI50 value (14.9 µM), without interference on cell growth until the highest tested concentration (500 µM). In contrast, HepG2 cancer cells reveled dose-dependent antiproliferative behavior, with GI50 = 97.5 µM. These results were better explored in the text.
  1. Figure 3: Attending to C and F graphs, Is not LC50 less than 500 micM?
    Response 9: Indeed, the LC50 values can not be seen on the graph since they were not found for any extracts and cell lines. The LC50 value represents the concentration that kills half of the cells (-50%) relative to the number at the start of treatment. This parameter considers the extracts cytotoxicity represented by the point -50 on the y axis of the graph, according to the formula: ([T - T0] / T0) × 100 = - 50. We explained the meaning of IC50, GI50 and LC50 parameters in item 3.5.
  1. Figure 4: Did authors not test on HepG2?
    Response 10: In fact we did not test the ROS assay on HepG2 cells. We chose the A549 cancer lung cell and its normal lung cell line (IMR90) as a control. This way we got comparable results from normal and cancer cells to better understanding and exploring their behavior in inducing or not ROS generation under these systems.
  1. Figure 4: Any significant difference? What are the meaning of the letters?
    Response 11: These letters represent the conclusion of statistical analyses by the test ANOVA, pós teste. Different letters in the same figure represent statistical difference among results and similar letters mean similar behavior. Thus, significant difference was found among the groups in ROS assay. First, the levels of ROS induced by H2O2 were higher than the negative control and they were similar to some treatment groups, such as 50 µM of compound 1, for A549 cells. Graphs with bars higher than negative control (different letter) mean that the compound induced intracellular ROS generation beyond the basal cell production. This behavior it was observed for both 1 and 2 compounds and for both A549 and IMR90 cell lines in different proportions, indicating pro-oxidant effects.

  1. Line 314: Why 48 h?
    Response 12: This kind of treatment is usually performed with 24, 48 or 72 hours [2]. Herein, we used 48 hours as published in our previous woks [3–6].

  1. Yang, J.; Yang, Y.; Wang, L.; Jin, Q.; Pan, M. Nobiletin selectively inhibits oral cancer cell growth by promoting apoptosis and DNA damage in vitro. Oral Surg. Oral Med. Oral Pathol. Oral Radiol. 2020, 130, 419–427, doi:10.1016/j.oooo.2020.06.020.
  2. Escher, G.B.; Marques, M.B.; do Carmo, M.A.V.; Azevedo, L.; Furtado, M.M.; Sant’Ana, A.S.; da Silva, M.C.; Genovese, M.I.; Wen, M.; Zhang, L.; et al. Clitoria ternatea L. petal bioactive compounds display antioxidant, antihemolytic and antihypertensive effects, inhibit α-amylase and α-glucosidase activities and reduce human LDL cholesterol and DNA induced oxidation. Food Res. Int. 2020, 128, 108763, doi:10.1016/j.foodres.2019.108763.
  3. Carmo, M.A.V. Do; Fidelis, M.; Pressete, C.G.; Marques, M.J.; Castro-Gamero, A.M.; Myoda, T.; Granato, D.; Azevedo, L. Hydroalcoholic Myrciaria dubia (camu-camu) seed extracts prevent chromosome damage and act as antioxidant and cytotoxic agents. Food Res. Int. 2019, 125, 108551, doi:10.1016/j.foodres.2019.108551.
  4. Maciel, L.G.; do Carmo, M.A.V.; Azevedo, L.; Daguer, H.; Molognoni, L.; de Almeida, M.M.; Granato, D.; Rosso, N.D. Hibiscus sabdariffa anthocyanins-rich extract: Chemical stability, in vitro antioxidant and antiproliferative activities. Food Chem. Toxicol. 2018, 113, 187–197, doi:10.1016/j.fct.2018.01.053.
  5. Migliorini, A.A.; Piroski, C.S.; Daniel, T.G.; Cruz, T.M.; Escher, G.B.; do Carmo, M.A.V.; Azevedo, L.; Marques, M.B.; Granato, D.; Rosso, N.D. Red Chicory ( Cichorium intybus ) Extract Rich in Anthocyanins : Chemical Stability , Antioxidant Activity , and Antiproliferative Activity In Vitro. J. Food Sci. 2019, doi:10.1111/1750-3841.14506.

  1. Line 359: Check reference style
    Response 13: Thanks for the correction. The correction was made in the text.
  1. Line 370: Check reference
    Response 14: Thanks for the correction. The correction was made in the text.
  1. Why did authors select those human cell lines?
    Response 15: These cells are recognized by the scientific community and we opted in using IMR90 (normal) and A549 (cancer) cells, once lung is an important system responsible for breathing and gas exchange. Additionally we selected another cancer cell line (HepG2), since metabolization is performed by liver. It is indispensable evaluate in vitro compounds behavior considering normal and cancer conditions
  1. Authors must improve introduction section (meaning of essential oils, sesquiterpenes, human cancer cells...)
    Response 16: Thanks for the suggestion. Our aim in this article was to present the isolation of the two sesquiterpenes, which are found only in this plant species. We made sure that all important bibliographic references related to these two compounds were cited in the text, presenting and discussing the most diverse issues related to the analysis of oils from Eugenia uniflora, and thus giving the reader all the security about the information.
  1. If they can perform some other antioxidant assay would be nice... (FRAP assay, superoxide radical...)
    Response 17: Thanks for the suggestion. Nonetheless, two antioxidant methods encompassing distinct mechanisms of action were used: single electron transfer (DPPH) and hydrogen atom transfer (ILP).

Round 2

Reviewer 1 Report

Dear Authors,

Thank you for taking into account most of issues presented in my previous review.

Reviewer 3 Report

Figure 4: Authors must explained the meaning of the letters in the caption of the figure.